# IgG-Based Bispecific Anti-CD95 Antibodies for the Treatment of B Cell-Derived Malignancies and Autoimmune Diseases

**DOI:** 10.3390/cancers14163941

**Published:** 2022-08-16

**Authors:** Sebastian Hörner, Moustafa Moustafa-Oglou, Karin Teppert, Ilona Hagelstein, Joseph Kauer, Martin Pflügler, Kristina Neumann, Hans-Georg Rammensee, Thomas Metz, Andreas Herrmann, Helmut R. Salih, Gundram Jung, Latifa Zekri

**Affiliations:** 1Department of Immunology, Institute for Cell Biology, Eberhard Karls University Tuebingen, German Cancer Consortium (DKTK), Partner Site Tuebingen, 72076 Tuebingen, Germany; 2Clinical Collaboration Unit Translational Immunology, German Cancer Consortium (DKTK), Department of Internal Medicine, University Hospital Tuebingen, 72076 Tuebingen, Germany; 3DFG Cluster of Excellence 2180 “Image-guided and Functional Instructed Tumor Therapy” (iFIT), Eberhard Karls University Tuebingen, 72076 Tuebingen, Germany; 4Charles River Discovery Research Services Germany GmbH, 79108 Freiburg, Germany; 5Baliopharm AG, 4051 Basel, Switzerland

**Keywords:** bispecific antibodies, lymphoma, autoimmune diseases, apoptosis, CD20, CD19, CD95

## Abstract

**Simple Summary:**

Therapeutic antibodies have become a crucial cornerstone of the standard therapy for lymphoma and autoimmune diseases. However, the respective target antigens are also expressed on healthy B cells resulting in unspecific effects. In this article, we present a novel approach to selectively induce apoptosis in lymphoma cells and autoreactive B cells that express the CD95 death receptor. Therefore, we developed an improved IgG-based bispecific antibody format with favorable production properties and pharmacokinetics for CD20- and CD19-directed induction of apoptosis via CD95. We could show that our bispecific anti-CD95 antibodies are very efficient in the depletion of malignant and autoreactive B cells in vitro and in vivo. Therefore, our antibodies could help to provide a more selective therapy for patients with B cell-derived malignancies and autoimmune diseases.

**Abstract:**

Antibodies against the B cell-specific antigens CD20 and CD19 have markedly improved the treatment of B cell-derived lymphoma and autoimmune diseases by depleting malignant and autoreactive B cells. However, since CD20 and CD19 are also expressed on healthy B cells, such antibodies lack disease specificity. Here, we optimize a previously developed concept that uses bispecific antibodies to induce apoptosis selectively in malignant and autoreactive B cells that express the death receptor CD95. We describe the development and characterization of bispecific antibodies with CD95xCD20 and CD95xCD19 specificity in a new IgG-based format. We could show that especially the CD95xCD20 antibody mediated a strong induction of apoptosis in malignant B cells in vitro. In vivo, the antibody was clearly superior to the previously used Fabsc format with identical specificities. In addition, both IgGsc antibodies depleted activated B cells in vitro, leading to a significant reduction in antibody production and cytokine secretion. The killing of resting B cells and hepatocytes that lack CD95 and CD20/CD19, respectively, was marginal. Thus, our results imply that bispecific anti-CD95 antibodies in the IgGsc format are an attractive tool for a more selective and efficient depletion of malignant as well as autoreactive B cells.

## 1. Introduction

B cells play a central role in the adaptive immune system. Differentiated into plasma cells, they produce antibodies. In addition, they act as antigen-presenting cells (APC) that produce a variety of cytokines and support T cell activation. Disrupting the finely balanced B cell homeostasis can lead to cancer or autoimmune diseases. Recombinant antibodies targeting B cell-associated antigens, such as CD20 and CD19, are successfully used for the treatment of B cell-derived malignancies and autoimmune diseases.

The anti-CD20 antibody Rituximab was the first recombinant antibody to be approved by the FDA in 1997 for the treatment of non-Hodgkin lymphoma (NHL) [1,2] and later for the treatment of chronic lymphocytic leukemia (CLL). Due to its profound potential to deplete CD20-positive B cells, it is now also successfully used to treat autoimmune diseases such as rheumatoid arthritis (RA) [3], Wegener’s granulomatosis [4], Sjögren syndrome [5], or multiple sclerosis (MS) [6]. Rituximab induces profound B cell depletion by mediating antibody-dependent cellular cytotoxicity (ADCC) and complement-dependent cytotoxicity (CDC). Although the antibody is relatively well tolerated, an increased risk of infections resulting from profound B cell depletion is a major concern [7]. Although CD19 appears to be more attractive than CD20 with respect to its broader expression on immature and differentiated B cells (plasma cells), clinical development of monospecific anti-CD19 antibodies is far less advanced [8]. Possibly due to its less pronounced expression, CD19 targeting was mainly used by T cell recruiting strategies, where limited numbers of target molecules per cell are less critical. Blinatumomab, a CD19xCD3 bispecific antibody (bsAb), which is used for the treatment of acute lymphoblastic leukemia (ALL), is the most prominent example [9,10]. Likewise, CD19 is successfully used as a target antigen for CAR T cells that are functionally closely related to bsAbs [11].

CD95 (Apo-1, Fas) is a member of the tumor necrosis factor superfamily (TNF) with an intracellular death domain for the induction of apoptosis [12]. In combination with its ligand FasL, it plays an important role in B and T cell homeostasis as these cell types increase CD95 expression in response to activation and become susceptible to the CD95-mediated apoptosis [13,14]. In addition, many cancer cells express CD95, although during disease progression it is frequently downregulated [15,16,17]. In fact, it had been the effective killing of lymphoma cells by an antibody (Apo-1) with—at that time—unknown specificity that allowed the identification of CD95 as a prototypical death receptor [18]. Thus, it appeared tempting to use CD95 agonists (either antibodies or FasL fusion proteins) to induce CD95-mediated apoptosis in cancer cells. However, it was soon recognized that the systemic administration of agonistic antibodies induces severe liver damage in mice [19,20,21]. This dramatically highlighted the need for more selective induction of apoptosis. In 2001, Jung et al. demonstrated that chemically hybridized bispecific F(ab)_2_ fragments with CD95xtarget specificity induce apoptosis selectively in tumor cells expressing the selected target antigen [22]. More recently, Nalivaiko et al. showed that recombinant bsAbs with CD95xCD20 specificity in the Fabsc-format were selectively inducing apoptosis not only in CD20 expressing tumor cells but also in normal, activated B cells expressing CD95 [23].

BsAbs that lack functional Fc parts such as Fabsc molecules or BiTEs (bispecific T cell engagers) suffer from a low serum half-life that may seriously limit therapeutic efficiency. Here, we introduce an IgG-based format (IgGsc) for target cell-restricted activation of CD95 [24]. We evaluated the ability of the IgGsc molecules to induce apoptosis in different lymphoma cell lines and in activated B cells in vitro. In addition, the CD95xCD20 IgGsc molecule was found to be superior to a Fabsc-molecule with identical specificity in an established lymphoma xenograft model.

## 2. Materials and Methods

### 2.1. Cells and Reagents

Daudi, Jurkat, SKW6.4, JY, C1R, Raji, and LX-1 cells were purchased from the American Type Culture Collection (ATCC, Manassas, VA, USA). Density-gradient centrifugation (Biocoll separating solution, Biochrom, Berlin, Germany) was used to isolate PBMCs from heparinized blood of healthy donors. All cells were cultured in RPMI 1640 compl. medium (Thermo Fisher Scientific, Darmstadt, Germany) containing 1x MEM-NEAA (Thermo Fisher Scientific), 10% FCS (Sigma-Aldrich, Hamburg, Germany), 100 U/mL penicillin, and 100 µg/mL streptomycin (Sigma-Aldrich), 1x sodium-pyruvate (Sigma-Aldrich) and 50 µM β-mercaptoethanol (Merck, Darmstadt, Germany). All cell lines were regularly tested for mycoplasma contamination and were cultured at 37 °C and 5% CO_2_.

The expression of CD20, CD19, and CD95 on different cell lines was determined by flow cytometry using QIFIKIT calibration beads (Agilent, Santa Clara, CA, USA) according to the manufacturer’s instructions. The hybridoma-derived antibodies 2H7, 4G7, and Apo-1 were used for quantification. Binding of bsAbs to Daudi and Jurkat cells was determined using flow cytometry. PE-conjugated goat anti-human F(ab)_2_ fragments were used to detect primary antibodies (Jackson ImmunoResearch, West Grove, PA, USA). Flow analysis was performed using the BD FACSCanto^TM^ II and BD FACSCalibur^TM^ systems (BD Biosciences, Heidelberg, Germany). Data were analyzed using FlowJo (FlowJo LLC, Ashland, OR, USA). EC_50_ values were calculated using GraphPad Prism9 (GraphPad Software, Inc., San Diego, CA, USA).

### 2.2. Generation and Purification of Recombinant Antibodies

The variable domains of the humanized Apo-1 antibody (EP2920210B1), the humanized 2H7 antibody (EP2920210B1), the 4G7 antibody (GenBank no.: AJ555479 and AJ555622), and the MOPC-21 antibody (GenBank no.: AAD15290.1 and AAA39002.1) were codon-optimized using the GeneArt GeneOptimizer tool for the transfection of CHO cells (Thermo Fisher Scientific). V_H_, V_L,_ and scFv sequences were synthesized de novo at GeneArt (Thermo Fisher Scientific). As previously described, the variable domains were inserted into a human IgGγ1sc backbone, which is designed to abolish FcR-binding and complement fixation [24]. IgGsc molecules were produced in the ExpiCHO^TM^ Expression System (Thermo Fisher Scientific) according to the manufacturer’s instructions and then purified by HiTrap^TM^ MabSelect^TM^ SuRe columns (Cytiva, Freiburg, Germany), before being subjected to preparative and analytical size exclusion chromatography (SEC) using HiLoad^TM^ 16/600 Superdex 200 pg and Superdex^TM^ 200 Increase 10/300 GL columns (Cytiva), respectively. Sodium dodecyl sulfate-polyacrylamide gel electrophoresis (SDS-PAGE) was performed as previously described [25]. The generation and purification of Fabsc molecules were described by Nalivaiko and colleagues [23].

### 2.3. Induction of Apoptosis and Caspase-3 Activation in Lymphoma Cells

The induction of apoptosis was evaluated by incubating 50,000 lymphoma cells/well (SKW6.4, JY, C1R, and Raji) in 96-well plates for 24 h with varying concentrations of different bsAbs before they were pulsed with 0.5 µCi/well ^3^H-thymidine (Hartmann Analytics, Braunschweig, Germany). After 20 h, cells were harvested on filter mats (Perkin Elmer, Waltham, MA, USA), and precipitated radioactivity was determined in a liquid scintillation counter (MicroBeta, Perkin Elmer).

SKW6.4 cells were also incubated with LX-1 hepatic stellate cells (50,000 cell/well each) for 20 h, before assessing viability with 7-AAD (BioLegend, San Diego, CA, USA) by flow cytometry. Absolute cell counts were calculated using equal numbers of latex beads (3 µm particle size, Sigma-Aldrich). LX-1 were distinguished from SKW6.4 using PE-EpCAM (clone 9C4, BioLegend).

For the determination of caspase-3 activity, 100,000 target cells were incubated for 20 h with 0.3 nM of bispecific antibody. Subsequent intracellular staining was performed using the Cytofix/Cytoperm buffer and Perm/wash solution (BD Biosciences) with an anti-active caspase-3 antibody (BD Biosciences) according to the manufacturer’s instructions. Samples were then analyzed by flow cytometry.

### 2.4. Depletion of Activated B Cells and Inhibition of IgG and Cytokine Production

For the bsAb-induced depletion of activated B cells, PBMCs from healthy donors (400,000 cells/well in 96-well plates) were stimulated with 0.1 µM ODN2006 (Miltenyi Biotec, Bergisch Gladbach, Germany) for 7 days [26]. Cells were then washed twice with DPBS before they were treated with 0.1 nM bsAb for 2 days. Lymphocytes were detected using CD4-FITC (clone OKT4), CD8-APC/Cy7 (clone SK1), CD19-PE/Cy7 (clone HIB19) or CD20-PE/Cy7 (clone 2H7), CD56-Brilliant Violet 421 (clone HCD56), CD69-PE (clone FN50) and CD95-APC (clone EH12.2H7). Cell viability was assessed using 7-AAD. All directly labeled antibodies were purchased from BioLegend. Equal numbers of latex beads (3 µm particle size, Sigma-Aldrich) were used to calculate absolute cell numbers.

Cytokines were measured using the Th1 LEGENDplex^TM^ multiplex kit (BioLegend) according to the manufacturer’s instructions. Only positive donors were used for analysis. The inhibition of IgG production of activated B cells after treatment with bsAbs was assessed by ELISA as previously described [23].

### 2.5. Animal Experiments

All experiments and protocols were approved by the animal welfare body at Charles River Discovery Research Services, Freiburg, Germany, and the local authorities, and were conducted in accordance with all applicable international, national, and local laws and guidelines (study number P500A4A). Only animals with unobjectionable health were selected to enter testing procedures.

To assess the activity of bsAbs against established tumors, 10^7^ SKW6.4 cells were injected subcutaneously into the left flank of female CB17 SCID mice. Mice were randomized if they bore a tumor of 50–250 mm^3^. The Fabsc molecule was administered intraperitoneal at 100 µg/mouse (twice daily for 10 days), while the IgGsc molecule was administered at 50 µg/mouse (on days 0 and 3). The animals were monitored at least once daily and were weighed three times a week. Blood was collected by retro-orbital sinus puncture and mice were sacrificed when their tumor had grown to a size limit of 2000 mm^3^. Tumor volumes were calculated according to the formula: tumor volume = (a × b^2^) × 0.5.

The serum concentrations of bsAb were measured by ELISA. An amount of 1 µg/mL CD95-Fc fusion protein was coated in 96-Well Half Area Microplates (Greiner Bio-One), before adding various dilutions of sera. Primary antibodies were detected using HRP-conjugated goat anti-human F(ab)2 specific antibodies (Jackson ImmunoResearch) and the TMB Peroxidase Substrate Kit (Seracare). The optical density at 450 nm was determined using a Spectra Max 340 (Molecular devices). Serum concentrations were interpolated by nonlinear regression using GraphPad Prism9.

### 2.6. Statistics

Data are presented as means ± SD or SEM as stated in the figure legends. Statistical significance was calculated with GraphPad Prism version 9.4 (GraphPad Software, San Diego, CA, USA) as indicated in the figure legends, with *p* < 0.05 considered statistically significant. ns *p* > 0.05, * *p* < 0.05, ** *p* < 0.01, *** *p* < 0.001, **** *p* < 0.0001.

## 3. Results

### 3.1. Construction of Improved Anti-CD95 bsAbs Targeting CD20 or CD19

Two different anti-CD95 antibodies in the IgGsc format were constructed (Figure 1a). The IgGsc format was originally published by Coloma and Morrison [27]. It contains two single-chain fragments variable (scFv) attached to the C-termini of an IgG1 antibody and was then further optimized in our group with a combination of multiple point mutations or deletions to eliminate the Fc receptor (FcR)-mediated multimerization of CD95 [24]. The F(ab)2 and the single-chain moiety of both molecules contained the CD95 agonist Apo-1 [18] and the anti-CD20 clone 2H7 [28] or the anti-CD19 clone 4G7 [29], respectively. This particular orientation was chosen because the Apo-1 antibody cannot be expressed as a single chain [23]. Analysis of both proteins by SDS-PAGE (Figure 1b) revealed the expected molecular weights of the heavy chain (75 kDa), the light chain (25 kDa), and the intact IgGsc molecule (200 kDa). Since the IgGsc format was previously designed for minimal aggregation, it lacked significant amounts of aggregates (less than 4%) as can be seen in gel filtration (Figure 1c). In addition, a CD95xMOPC control was generated to assess CD20- and CD19-independent effects (Appendix A).

The binding of the molecules to CD95^+^ Jurkat cells (CD20^−^ and CD19^−^, Appendix A) was evaluated by flow cytometry (Figure 1d) and revealed a binding affinity in a low nanomolar range (EC_50_ = 0.2 nM). Binding to CD20 and CD19 was assessed using double positive Daudi cells (Appendix A). Although the expression of the CD20 antigen was higher than that of CD19, the binding affinity of the CD20 binding moiety was rather low. Saturation could not be reached for concentrations up to 900 nM (Figure 1e). Obviously, conversion of the anti-CD20 clone into an scFv resulted in a significant loss of affinity compared to the parental antibody as previously described by Nalivaiko and colleagues [23]. In contrast, the binding affinity to CD19 revealed an EC_50_ value of approximately 7.5 nM (Figure 1f).

### 3.2. In Vitro Activity against Malignant B Cells

The ability of the CD95xCD20 and CD95xCD19 constructs to induce apoptosis was assessed using the B cell lymphoma cell lines SKW6.4, JY, C1R, and Raji. All cells tested positive for CD95, expressing at least 90,000 molecules per cell (Figure 2a and Table 1). The expression of the target molecules CD20 and CD19 revealed significant differences. CD20 expression was 3–10-fold higher on all cell lines tested. Only Raji cells expressed more than 20,000 CD19 molecules per cell, while expression of CD20 was higher than 190,000 molecules per cell on all lymphoma cell lines tested in this work.

We further assessed the ability of our bsAbs to inhibit cell proliferation in a ^3^H-thymidine-based proliferation inhibition assay. SKW6.4, JY, and C1R were sensitive to antibody treatment, resulting in a concentration-dependent inhibition of proliferation, while Raji cells were CD95-resistant despite expressing sufficient levels of CD95, CD20, and CD19 (Figure 2b). The CD95xCD20 bsAb showed the most pronounced cell reduction resulting in IC_50_ values between 24 and 35 pM (Table 1). In contrast, treatment with the CD95xCD19 bsAb exhibited only moderate inhibition of proliferation. This result was surprising given the higher binding affinity of the anti-CD19 clone 4G7 compared to the anti-CD20 clone 2H7 (Figure 1e,f). Our results suggest that the significantly higher expression of CD20 on lymphoma cell lines seems to be an important prerequisite for triggering efficient CD95 signaling (Figure 2a). The CD95xMOPC molecule also showed moderate inhibition of cell proliferation, but only at high concentrations, indicating that the observed effect is highly dependent on the presence of the respective target antigen.

To further confirm that the observed inhibition of proliferation is due to apoptosis, an intracellular staining of active caspase-3 was performed using flow cytometry. The results depicted in Figure 2c demonstrated that upon treatment with CD95xCD20 bsAb a significantly higher active caspase-3 staining was observed. In contrast, no significant increase was observed with CD95xCD19 and the control bsAb.

### 3.3. In Vivo Activity against Malignant B Cells

Next, we decided to compare the newly generated CD95xCD20 molecule in the IgGsc format to the previously generated Fabsc molecule by Nalivaiko and colleagues in vitro and in vivo (Figure 3a) [23]. CD19 bsAbs were not included in in vivo assays as they showed no significant therapeutic effects on lymphoma cell lines. In vitro, both CD95xCD20 antibody formats induced similar killing of SKW6.4 cells (Figure 3b).

In vivo antitumor activity of CD95xCD20 was examined using established SKW6.4 tumor models in CB17 SCID mice. Antibodies in the Fabsc format were shown to have a very short serum half-life, which is greatly increased in IgGsc molecules, as the latter contains a functional CH3 that allows binding to FcRn and recycling of the molecule (Figure 3d) [23,24]. Accordingly, the IgGsc molecule was dosed twice on days 0 and 3 (50 µg per mouse, 100 µg total), while the Fabsc molecule was injected twice daily for 10 days (100 µg per mouse, 2 mg total) to compensate for the lower serum half-life. Both molecules had no side effects as the mice were monitored three times a week and no loss in weight was observed (Figure 3c). The Fabsc showed tumor regression until day ~15, but once the antibody injection was stopped the tumors started to regrow rapidly. In marked contrast, the IgGsc construct showed tumor regression until day ~40, despite a 20-fold reduced total dose compared to the Fabsc (Figure 3e). Consequentially, the IgGsc molecule resulted in improved overall survival of the mice (Figure 3f).

### 3.4. Depletion of Activated B Cells and Reduction of IgG and Cytokine Levels

Next, we assessed the activity of the IgGsc constructs to induce apoptosis in activated B cells. The intention was to mimic the situation in patients with autoimmune diseases, where activated B cells produce autoantibodies and cytokines or act as APCs that stimulate T cells against self-antigens. Unlike CD19, CD20 is not expressed on antibody-producing plasma cells. Therefore, the CD95xCD19 construct was again included in these experiments, even if it showed only minor activity against lymphoma cells. Peripheral blood mononuclear cells (PBMCs) from healthy individuals were stimulated for 7 days with toll-like-receptor 9 (TLR9) agonistic CpG oligodeoxynucleotides (ODN2006), before being treated with the bsAbs for 2 days [26]. ODN2006 represents a B-class ODN for the activation of B cells, leading to the upregulation of CD20, CD19, and especially CD95 (Appendix A). The expression of CD20 (~124,000 molecules per cell) and CD19 (~22,000 molecules per cell) on activated B cells was comparable to lymphoma cell lines (see also Figure 2a and Table 1).

The bispecific anti-CD95 antibodies induced a pronounced target cell-mediated depletion of activated B cells with comparable efficacy for the CD95xCD20 and CD95xCD19 construct (Figure 4a). The depletion of activated B cells also resulted in reduced IgG and cytokine (IL-6 and IL-10) levels in the supernatant (Figure 4c,d). Both cytokines are associated with different autoimmune diseases [30]. IL-2 and TNFα could not be detected. IFNγ secretion appeared to be reduced after antibody treatment but was highly donor-dependent and the effect was overall not significant (data not shown). The secretion of different cytokines most probably also led to minor activation of T and NK cells, resulting in “bystander killing” by cross-linking with CD20- and CD19-expressing B cells (Figure 4a and Appendix A). The killing of B cells within resting PBMC preparations was much weaker, highlighting the specificity for CD95-expressing activated lymphocytes (Figure 4b).

### 3.5. Anti-CD95 bsAbs in the IgGsc Format Do Not Induce Apoptosis in Hepatocytes

Systemic administration of anti-CD95 antibodies can lead to fulminant hepatitis. This depends, among other reasons, on the antibody clone used and whether it stimulates type I apoptosis (mitochondria-independent) or type II apoptosis (mitochondria-dependent) [19,31]. SKW6.4 and activated lymphocytes are described as type I cells, while hepatocytes are considered type II cells.

To ensure that our anti-CD95 clone Apo-1 does not induce apoptosis in hepatocytes, we co-incubated SKW6.4 and the hepatic stellate cell line LX-1 with our bsAbs (Figure 5). This resulted in a target cell-mediated depletion of SKW6.4 lymphoma cells (CD20^+^/CD19^+^/CD95^+^), while LX-1 cells (CD20^−^/CD19^−^/CD95^+^) were unaffected (Appendix A).

## 4. Discussion

The selective activation of the CD95 death receptor by bsAbs is a very attractive strategy for depletion of undesired B cells that express CD95. To this end, we developed two antibodies in the IgGsc format targeting the B cell-restricted antigens CD20 and CD19 for target cell-mediated induction of apoptosis. The IgGsc format was chosen for its favorable pharmacokinetic and producibility [24].

CD20 is a rather small antigen (33–35 kDa) and the antibody clone 2H7 binds in close proximity to the cellular membrane to a similar epitope as does rituximab [32]. Herrmann and colleagues reported that anti-CD95 bsAbs induce apoptosis rather in trans than in cis configuration [33]. Therefore, the architecture of CD20 might facilitate bicellular binding of bsAbs and thus creates optimal conditions for CD95 clustering. The combination with its high expression levels on malignant and activated B cells make CD20 a very promising candidate for targeted induction of apoptosis. Our CD95xCD20 bsAb revealed IC_50_ values as low as 25 pM, which was remarkable considering the rather low affinity of the anti-CD20 clone 2H7 as a single chain. The construction of the molecule in a “reversed orientation”, with an N-terminal CD20 binder, was not possible as the anti-CD95 clone Apo-1 cannot be expressed as a single chain as some antibodies tend to show high levels of aggregation in this confirmation [23,34]. CD19 expression was much weaker on malignant cells, reducing apoptotic effects.

In a recent publication, we could show that the IgGsc format has a much better pharmacokinetic as compared to our previous Fabsc format [24]. Therefore, we decided to compare both formats with CD95xCD20 specificity in CB17 SCID mice bearing established SKW6.4 tumors. To compensate for Fabsc’s lower serum half-life, its total dose was 20-fold higher compared to the IgGsc. Nevertheless, the anti-tumor efficacy of the IgGsc was clearly superior due to its improved serum half-life and, as demonstrated by Zekri et al., due to sustained tumor localization.

A limiting factor for the treatment of lymphomas with our bispecific antibodies of the described kind is loss of CD95 expression or sensitivity [16,17]. The Burkitt lymphoma cell line Raji for example showed high expression levels of CD95 but was still resistant to CD95-mediated apoptosis. The combination with the Bcl-2 inhibitor Venetoclax or Doxorubicin could help restore CD95 sensitivity.

In contrast to malignant cells, normal B cells acquire CD95 expression and sensitivity during the activation [13,14]. This plays a crucial role in regulating the homeostasis of B cell activation, demonstrated by the fact that disorders of CD95-mediated apoptosis can lead to the autoimmune lymphoproliferative syndrome (ALPS) [35,36]. Therefore, bsAbs with CD95xtarget-specificity could be successfully used for the treatment of B cell-mediated autoimmune diseases as these cells should not escape CD95-mediated apoptosis as easily as malignant cells.

Our study confirmed that B cells activated for 7 days are highly sensitive to CD95 signaling. The anti-CD95 bsAbs induced profound B cell depletion and subsequent reduction in IgG and cytokine levels. In contrast to lymphoma cells, the CD95xCD19 construct was also very effective at depleting activated B cells. Since the expression of CD19 on malignant and normal B cells is comparable, this can possibly be explained by a higher sensitivity of normal lymphocytes to CD95-induced cell death and/or the expression of this antigen on antibody-producing plasma cells [37,38]. However, autoantibody production is not the only problem in B cell-mediated autoimmune diseases, as B cell-derived cytokines such as IL-6 can support autoimmunity. Indeed, it has been demonstrated that B cell-derived IL-6 plays an important role in experimental autoimmune encephalomyelitis (EAE) and MS [39]. The secretion of cytokines most likely led to the activation of other lymphocytes in our experiments and explains the “bystander killing” of T and NK cells. In any case, our antibodies showed only minor depletion of resting PBMCs, highlighting to specificity for activated lymphocytes.

Activation of the CD95 death receptor depends on the clustering and immobilization [21]. Immobilization enhances the activity of soluble FasL by several orders of magnitude, but soluble FasL can still induce apoptosis. In principle, our anti-CD95 bsAbs, by inducing “bivalent CD95 stimulation”, might also be able to trigger the CD95 death receptor without immobilization in higher concentrations. In patients, this could potentially lead to hepatic damage since hepatocytes are very sensitive to CD95 activation. However, several publications indicate that depending on the antibody clone apoptosis is rather induced in activated lymphocytes (type 1 cells) than in hepatocytes (type II cells) [19,31]. Indeed, we did not observe apoptosis of hepatic stellate cells in the presence of our anti-CD95 (clone Apo-1) bsAbs. Thus, in conclusion, we believe that our novel CD95xCD20 and CD95xCD19 constructs are attractive reagents for treatment of B cell-derived malignancies and autoimmune diseases by targeted induction of apoptosis.

## 5. Conclusions

In summary, bispecific anti-CD95 antibodies are promising candidates for a more specific depletion of malignant and autoreactive B cells and hold promise to improve the safety and efficacy compared to previously established antibody therapies.

## 6. Patents

AH is listed as an inventor of the patent application “recombinant bispecific antibody binding to CD20 and CD95”, EP2920210B1, applicant Baliopharm AG, Basel, Switzerland.

## Figures and Tables

**Figure 1 cancers-14-03941-f001:**
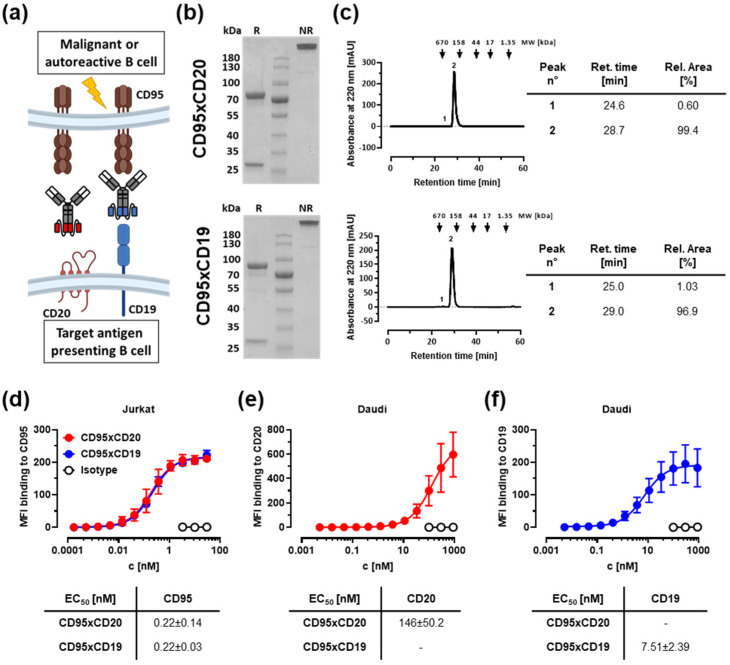
Characterization of improved anti-CD95 bsAbs targeting CD20 or CD19. (**a**) Schematic representation of how the bsAbs induce a target-mediated clustering of the Fas receptor and ultimately lead to apoptosis of the target cell. Created with BioRender.com (accessed on 18 May 2022). (**b**) SDS-PAGE of the bispecific CD95xCD20 (top) and CD95xCD19 (bottom) molecules. R: reduced; NR: non-reduced. (**c**) Both antibodies were subjected to analytical size exclusion chromatography (SEC). The corresponding analysis is presented in the tables on the right. (**d**–**f**) Binding to CD95-expressing Jurkats (**d**) and Daudi cells expressing CD20 (**e**) and CD19 (**f**) was assessed by flow cytometry. Mean ± SD, *n* = 3.

**Figure 2 cancers-14-03941-f002:**
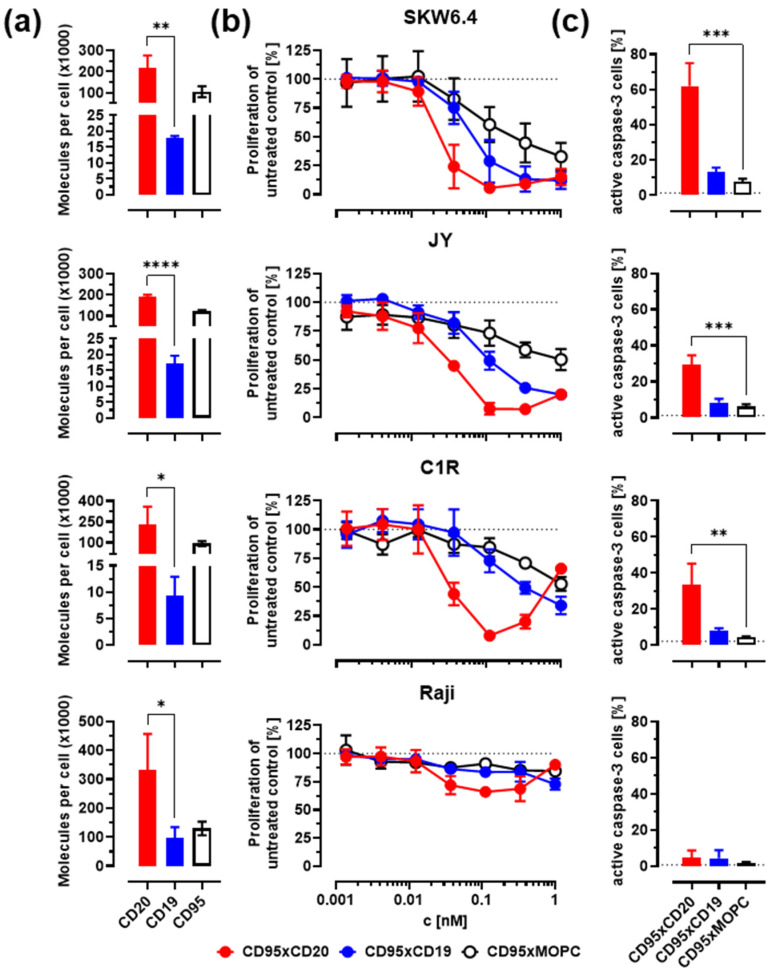
Bispecific CD95 antibodies induce depletion of lymphoma cells via apoptosis. The target antigen-mediated induction of apoptosis was evaluated on SKW6.4, JY, C1R, and Raji cells. (**a**) The antigen density of CD95, CD20, and CD19 on the cell surface of different cell lines was calculated using the flow cytometry-based Qifikit system. Statistics were calculated using an unpaired *t*-test, CD20 vs. CD19 expression. (**b**) Different lymphoma cell lines were incubated for 48 h with different concentrations of bispecific antibodies. Inhibition of cell proliferation was evaluated using a ^3^H-thymidine uptake. (**c**) Induction of apoptosis with the indicated bsAbs (0.3 nM) was verified by intracellular active caspase-3 staining after 20 h using flow cytometry. Statistics were calculated with one-way ANOVA. Treatment versus isotype control. Mean ± SD, *n* = 3. * *p* < 0.05, ** *p* < 0.01, *** *p* < 0.001, **** *p* < 0.0001. Further analysis of (**a**,**b**) is also presented in Table 1.

**Figure 3 cancers-14-03941-f003:**
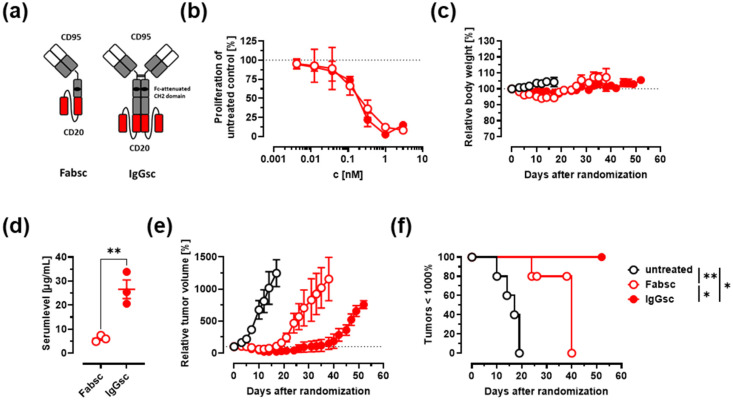
Antitumor activity of the bispecific Fabsc and IgGsc molecules (CD95xCD20) in immunodeficient mice. (**a**) Schematic representation of the Fabsc (left) and the IgGsc format (right). (**b**) The lymphoma cell line SKW6.4 was incubated for 48 h with different concentrations of bispecific antibodies. Inhibition of proliferation was measured using a ^3^H-thymidine uptake. Mean ± SD, *n* = 3. (**c**–**f**) 10^7^ SKW6.4 cells were injected s.c. in CB17 SCID mice, and antibody treatment was started when tumors reached a volume of 50–250 mm^3^. The Fabsc molecule was given twice daily at 100 µg/mouse for 10 days (total dose 2 mg/mouse), while the IgGsc molecule was only dosed twice (on day 0 and day 3) at 50 µg/mouse (total dose 100 µg/mouse). Mean ± SEM, 3–5 mice per group. (**c**) The body weight of the mice was controlled 3 times per week. (**d**) Serum concentrations of the Fabsc and the IgGsc molecule 1 h after injection of the dose on day 10 or on day 3, respectively. Unpaired *t*-test was used for statistical analysis. (**e**) The relative tumor volume over time and (**f**) Kaplan–Meier plot of the experiment shown in (**e**). Log-rank (Mantel–Cox) test for pairwise comparisons was used for statistical analysis. * *p* < 0.05, ** *p* < 0.01.

**Figure 4 cancers-14-03941-f004:**
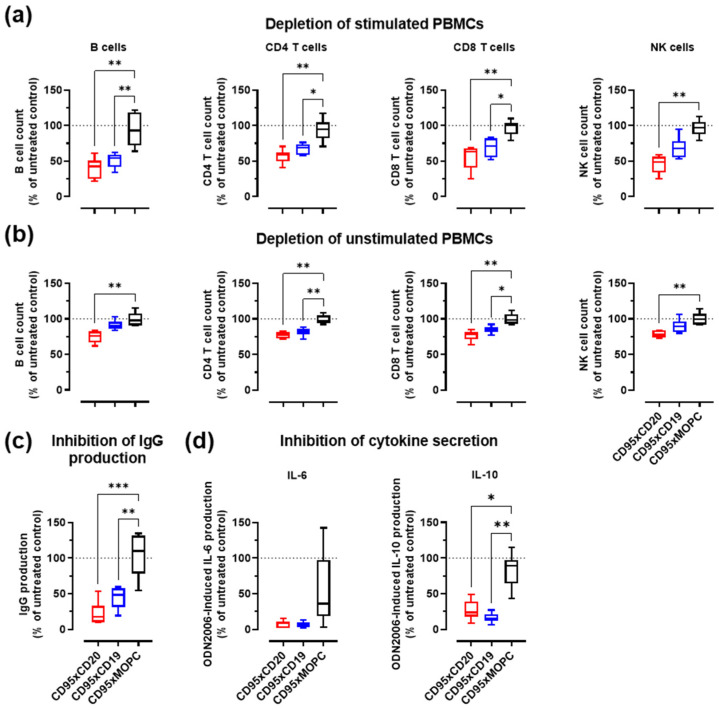
Bispecific CD95 antibodies induce depletion of activated B cells and reduce the production of IgG antibodies and cytokines. CD95xCD20 in red, CD95xCD19 in blue and CD95xMOPC in white. (**a**) Human PBMCs were activated with 0.1 µM ODN2006 for 7 days before they were incubated for 2 days with 0.1 nM of bispecific antibody. The viability of B cells, T cells, and NK cells was then analyzed by flow cytometry and normalized to the untreated controls. (**b**) Unstimulated human PBMCs were treated for 2 days with 0.1 nM of bispecific antibody and then analyzed as in (**a**). (**c**) The supernatant from (**a**) was analyzed for inhibition of IgG production and (**d**) for the secretion of IL-6 and IL-10 from activated B cells. Boxplot and whiskers from six different donors. Statistics were calculated with one-way ANOVA. Treatment versus isotype control. * *p* < 0.05, ** *p* < 0.01, *** *p* < 0.001.

**Figure 5 cancers-14-03941-f005:**
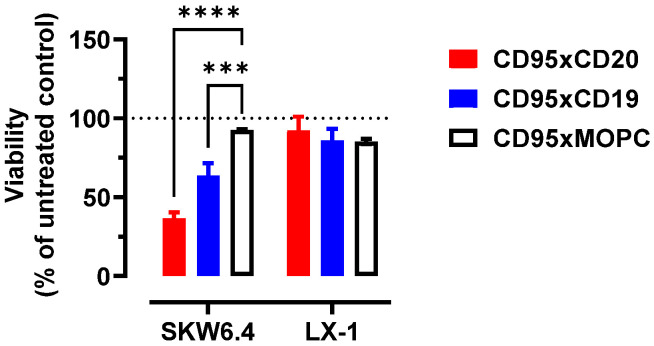
Bispecific anti-CD95 antibodies induce apoptosis in activated lymphocytes (type I cells) but not in hepatocytes (type II cells). SKW6.4 lymphoma cells and LX-1 hepatic stellate cells were co-cultured for 20 h before the depletion of both cell populations was determined by flow cytometry. Statistics were calculated with two-way ANOVA. Treatment versus isotype control. Mean ± SD, *n* = 3. *** *p* < 0.001, **** *p* < 0.0001.

**Table 1 cancers-14-03941-t001:** Antigen expression on lymphoma cell lines and IC_50_ values of bispecific antibodies. The number of CD20, CD19, and CD95 molecules expressed on different lymphoma cell lines (see also Figure 2a) as well as the absolute IC_50_ values (pM) of different bispecific antibodies (see also Figure 2b). Mean ± SD, *n* = 3.

	Molecules/Cell (×1000)			CD95xCD20	CD95xCD19	CD95xMOPC
Cell Line	CD20	CD19	CD95	abs. IC_50_	abs. IC_50_	abs. IC_50_
SKW6.4	218 ± 58	18 ± 1	105 ± 26	24 ± 11	69 ± 29	481 ± 571
JY	190 ± 10	17 ± 3	122 ± 5	35 ± 1	107 ± 35	-
C1R	233 ± 124	9 ± 4	93 ± 17	32 ± 7	306 ± 154	-
Raji	332 ± 125	99 ± 36	130 ± 24	-	-	-

## Data Availability

The data presented in this study are available on request from the corresponding author.

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
