# Peer review of "IgG-Based Bispecific Anti-CD95 Antibodies for the Treatment of B Cell-Derived Malignancies and Autoimmune Diseases"

_cancers, 2022, doi:10.3390/cancers14163941_

Round 1

Reviewer 1 Report

This manuscript is well-written, and the topic is interesting.

1.  Did you analyze other cytokines  (IL-2, IL-4, TNF-alpha, interferon-r, and TGF-b) after treatment of anti-CD95 bispecific antibodies? what about the effect on T cells? ( T cells also express CD95).

2. IL-10 is known to be released by regulatory B cells. Do you think that this is a bystander effect?

3. Did you compare the anti-tumor effect of the anti-CD20-CD95 bispecific antibody with rituximab in vitro and in vivo?

4. Do you have a PK-data comparing with FabSc?

5.  what about in vitro cytotoxicities of these anti-CD19-CD95  and anti-CD20-CD95 bispecific antibodies compared with the corresponding FabSc?

6.  Did you test the anti-tumor efficacy of these CD95 bispecific antibodies with other representative leukemia or lymphoma cell lines?

7. Do you have in vivo data of anti-CD19-CD95 bispecific antibody?

Reviewer 2 Report

The authors summarize development of two antibodies in the IgGsc format targeting the B cell-restricted antigens CD20 and CD19 for target cell-mediated induction of apoptosis Overall the article is interesting, well-written and is worthy of publishing in Cancers. 

The only other modifications I see as appropriate would be a) long-term stability of this BiAbs b) PK study results of this BisAbs c) a semi-quantitative assessment of developability and scalability of this mAb production.
